# Effect of Intermittent Fasting on Reproductive Hormone Levels in Females and Males: A Review of Human Trials

**DOI:** 10.3390/nu14112343

**Published:** 2022-06-03

**Authors:** Sofia Cienfuegos, Sarah Corapi, Kelsey Gabel, Mark Ezpeleta, Faiza Kalam, Shuhao Lin, Vasiliki Pavlou, Krista A. Varady

**Affiliations:** Department of Kinesiology and Nutrition, University of Illinois at Chicago, Chicago, IL 60647, USA; scienf2@uic.edu (S.C.); scorap2@uic.edu (S.C.); kdipma2@uic.edu (K.G.); mezpel2@uic.edu (M.E.); fkalam2@uic.edu (F.K.); slin89@uic.edu (S.L.); pavlou2@uic.edu (V.P.)

**Keywords:** intermittent fasting, time-restricted eating, weight loss, reproductive hormones, estrogen, testosterone, androgens, gonadotropins, prolactin, males, females

## Abstract

Intermittent fasting is a popular diet for weight loss, but concerns have been raised regarding the effects of fasting on the reproductive health of women and men. Accordingly, we conducted this literature review to clarify the effects of fasting on reproductive hormone levels in humans. Our results suggest that intermittent fasting decreases androgen markers (i.e., testosterone and the free androgen index (FAI)) while increasing sex hormone-binding globulin (SHBG) levels in premenopausal females with obesity. This effect was more likely to occur when food consumption was confined to earlier in the day (eating all food before 4 pm). In contrast, fasting did not have any effect on estrogen, gonadotropins, or prolactin levels in women. As for men, intermittent fasting reduced testosterone levels in lean, physically active, young males, but it did not affect SHBG concentrations. Interestingly, muscle mass and muscular strength were not negatively affected by these reductions in testosterone. In interpreting these findings, it is important to note that very few studies have been conducted on this topic. Thus, it is difficult to draw solid conclusions at present. From the limited data presented here, it is possible that intermittent fasting may decrease androgen markers in both genders. If this is the case, these results would have varied health implications. On the one hand, fasting may prove to be a valuable tool for treating hyperandrogenism in females with polycystic ovarian syndrome (PCOS) by improving menstruation and fertility. On the other hand, fasting may be shown to decrease androgens among males, which could negatively affect metabolic health and libido. More research is warranted to confirm these preliminary findings.

## 1. Introduction

Intermittent fasting has gained tremendous popularity over the last decade as a weight loss regimen [1,2]. Intermittent fasting is an umbrella term for three different diets: alternate day fasting (ADF), the 5:2 diet, and time-restricted eating (TRE). ADF involves a “feast day”, where individuals eat ad libitum, alternated with a “fast day”, where participants can choose to consume only water or consume ∼25% of energy needs. The 5:2 diet, on the other hand, is a modified version of ADF that involves 5 feast days and 2 fast days per week. Finally, TRE involves confining the eating window to a specified number of hours per day (usually 4 to 10 h) and fasting with zero-calorie beverages for the remaining hours [1,2]. Accumulating evidence indicates that these various intermittent fasting regimens are effective for decreasing body weight and improving insulin sensitivity, blood pressure, and markers of oxidative stress in adults with obesity [1,3,4,5].

Nevertheless, concerns have been raised regarding the effects of fasting on the reproductive health of women and men. For instance, some women are skeptical about starting intermittent fasting because they believe it may negatively affect levels of estrogen and other reproductive hormones, leading to menstrual cycle irregularities and fertility issues. These concerns largely stem from the findings of one rodent study by Kumar et al. [6]. This study [6] is heavily referenced in the popular media and is often used as evidence to show that fasting is disruptive to female reproduction. In their experiment [6], young rats underwent 24 h of water fasting every other day for 12 weeks. By the end of the trial, serum estradiol increased, while luteinizing hormone (LH) levels decreased versus ad libitum fed controls. Negative changes in estrous cyclicity (i.e., menstrual cycle) were also observed. While these findings are indeed concerning, it should be noted that the female rats were very young (3 months old), which corresponds to a human aged 9 years old [7]. Intermittent fasting is not recommended for children under the age of 12, since it has the potential to negatively impact puberty and growth [2]. Thus, data from the Kumar et al. study [6], though valuable to the field, do not provide evidence for how fasting may impact reproductive hormone levels in adult females.

Some men are also wary about starting intermittent fasting, as they believe it may alter testosterone levels. These concerns come from a study by Moro et al. [8], where young lean males participated in an 8 h TRE intervention (all food consumed between 12 to 8 p.m.) combined with resistance training. After 8 weeks of intervention, reductions in free and total testosterone concentrations were observed. Interestingly, the decreases in the anabolic hormone, testosterone, did not lead to any deleterious body composition changes or compromises in muscular strength. These findings highlight the need for more research to be conducted in this field.

To date, the effects of intermittent fasting on reproductive hormone levels in humans remain largely unknown, since no reviews of the literature have been performed. In view of this research gap, we conducted this review to further clarify the effects of intermittent fasting on sex hormone levels in adult female and male participants.

## 2. Methods—Human Trial Selection

A PubMed, CINAHL, EBSCOhost, and Cochrane Library search was conducted using the following key words: “intermittent fasting”, “meal timing”, “meal frequency”, “delayed meal”, “intermittent energy restriction”, “intermittent calorie restriction”, “alternate day fasting”, “5:2 diet”, “time restricted eating”, “time restricted feeding”, “testosterone”, “estrogen”, “estradiol”, “progesterone”, “prolactin”, “DHEA”, “follicle-stimulating hormone”, “luteinizing hormone”, “androgen”, “sex hormone binding globulin”, “sex hormone”, and “free androgen index”. The inclusion criteria for research articles were as follows: (1) randomized controlled trials and nonrandomized trials; (2) adult male and female participants (>18 years); (3) endpoints that included changes in body weight and at least one sex hormone. The following exclusion criteria were applied: (1) cohort and observational studies; (2) fasting performed as a religious practice (e.g., Ramadan or Seventh Day Adventist); (3) trial durations of less than 1 week; (4) studies that combined data for males and females. Our search retrieved 5 human trials of TRE, [8,9,10,11,12] one human trial of the 5:2 diet, [13], and one study [14] that examined the effect of meal timing on reproductive hormone concentrations. The results from these trials were separated for females (Table 1) and males (Table 2). We were not able to find any studies on ADF that examined sex hormones; therefore, this form of fasting was not included in this review.

### 2.1. Females: Effects of Intermittent Fasting on Reproductive Hormone Concentrations

#### 2.1.1. Estradiol

Estradiol is a naturally occurring form of estrogen [15]. It is the main female reproductive hormone and is involved in the development and maintenance of female reproductive tissues and the regulation of the menstrual cycle [15]. Women with overweight and obesity have higher levels of estrogens relative to their normal-weight counterparts [16]. An increase in estrogens derived from excess adiposity is linked to adverse health outcomes such as polycystic ovarian syndrome (PCOS), anovulation, and increased breast cancer risk [17,18,19]. Weight loss interventions have been shown to reduce estrogen levels among females with obesity [20].

To date, only one trial [14] has measured the effect of meal timing on estradiol levels in women (Table 1). Jakubowicz et al. [14] compared the effect of eating >50% of calories at dinner versus eating >50% of calories at breakfast in females with PCOS. After 12 weeks, estradiol significantly increased among females with PCOS when participants ate >50% of daily calories at dinner [14]. This result suggests that eating a large amount of food later in the day may augment estrogen levels in women with PCOS. Elevated serum androgens are converted to estrogens in adipose tissue, leading to excess estrogen production in females with PCOS and obesity. The increased estrogen production impairs the function of the hypothalamic–pituitary–gonadal axis. Thus, excess of estrogen and androgen are the primary causes of anovulation in PCOS patients [19]. Taken together, shifting calorie intake to earlier in the day may be preferred for females with PCOS to avoid further increases in estrogen levels.

#### 2.1.2. Androgens (Testosterone, DHEA-S, Androstenedione, and FAI)

Hyperandrogenism is a medical condition characterized by high levels of androgens, i.e., testosterone, dehydroepiandrosterone-sulfate (DHEA-S), and androstenedione. In females, hyperandrogenism is portrayed by hirsutism (i.e., excessive hair growth), seborrhea (scaly patches on the body and scalp), and disorders in the menstrual cycle [21]. Studies have reported that hyperandrogenism promotes insulin resistance and visceral adiposity among females by decreasing whole-body glucose uptake [22,23]. In females with PCOS and obesity, weight loss has been shown to decrease testosterone and androstenedione levels while increasing SHBG concentrations [24,25]. Abnormal androgen status is measured by the free androgen index (FAI). The FAI is a ratio that is calculated by dividing total testosterone by sex hormone-binding globulin (SHBG) and then multiplying by 100.

Three studies [9,13,14] have examined the effect of intermittent fasting on androgen markers in females (Table 1). In a study by Harvie et al. [13], premenopausal women with obesity followed a 5:2 diet where they fasted with 500 kcal two days per week. After 24 weeks of 5:2, the FAI significantly decreased, with a 7% weight loss versus baseline. DHEA-S, testosterone, and androstenedione, on the other hand, remained unchanged [13]. In addition, two studies [9,14] examined the effects of fasting on androgens in women with PCOS. Li et al. [9] conducted an 8 h TRE trial, where young women with PCOS and obesity ate all of their energy needs early in the day (between 8 am and 4 pm) and fasted with water for the rest of the day for 5 weeks. This early 8 h TRE intervention significantly decreased body weight by 2%, along with FAI and total testosterone levels. Complementary to these findings, Jakubowicz et al. [14] compared the effect of eating >50% of calories at dinner versus eating >50% of calories at breakfast in females with PCOS. The results showed that FAI, DHEA-S, and androstenedione decreased significantly in the breakfast group relative to the dinner group. These changes occurred without weight loss. In these trials [9,13,14], changes in androgen markers were also accompanied by reductions in body weight, inflammation, and insulin resistance, further illustrating the link between hyperandrogenism and metabolic disturbances. Altogether, these findings suggest that fasting can significantly decrease androgen markers in premenopausal females and those with PCOS, especially when calories are consumed earlier in the day.

#### 2.1.3. SHBG

Sex hormone-binding globulin (SHBG) is a hepatocyte-produced glycoprotein. The principal function of SHBG is to transport testosterone and estradiol to target tissues. Thus, the bioavailability of these reproductive hormones is influenced by circulating levels of SHBG [26]. Observational studies show that low levels of SHBG are associated with an increased incidence in insulin resistance and type 2 diabetes, independent of sex hormone concentrations [27]. Moreover, low levels of SHBG are frequently found in females with PCOS and contribute to hyperandrogenic symptoms such as hirsutism, acne, and androgenic alopecia [28,29,30]. Weight loss and has been shown to increase SHBG and improve insulin sensitivity in women with obesity and PCOS [31,32].

Three trials [9,13,14] examined how fasting impacts circulating concentrations of SHBG in females (Table 1). Li et al. [9] showed that concentrations of SHBG significantly increased, with 2% weight loss after 5 weeks of 8 h TRE in females with PCOS. The study by Harvie et al. [13] also demonstrated a significant increase in circulating SHBG levels, with a 7% weight loss after 24 weeks of the 5:2 diet in premenopausal women with obesity. Moreover, in the study by Jakubowicz et al. [14], SHBG levels increased when females with PCOS ate >50% of their calories in the morning compared to eating those calories in the evening. Interestingly, these improvements in SHBG occurred despite no change in body weight [14]. These preliminary findings suggest that intermittent fasting regimens may produce beneficial increases in SHBG concentrations in premenopausal females and those with PCOS, particularly when most of the food is consumed in the morning or afternoon.

#### 2.1.4. Gonadotropins

Gonadotropins are peptide hormones that regulate ovarian function and are essential for normal growth, sexual development, and reproduction [33]. Human gonadotropins include follicle-stimulating hormone (FSH) and luteinizing hormone (LH), which are made in the pituitary gland [33]. Previous studies have shown that weight loss reduces the LH/FSH ratio, with an FSH predominance favoring the maturation of the ovarian follicle (folliculogenesis) [34].

Changes in gonadotropins during fasting have only been assessed in one clinical trial to date [9] (Table 1). In this trial by Li et al. [9], young women with obesity and PCOS followed an early 8 h TRE regimen for 5 weeks. At the conclusion of the study, LH and FSH remained unchanged, even though these participants lost a small amount of weight (2% from baseline) and fat mass (2.4 kg from baseline). Visceral fat also decreased, but the skeletal muscle remained unchanged. It is possible that the degree of weight loss and fat mass loss was not sufficient to modulate the LH and FSH concentrations.

#### 2.1.5. Prolactin

Prolactin is a hormone responsible for milk production and mammary gland development [35]. Accumulating evidence suggests that weight loss via dietary interventions does not significantly affect prolactin levels [36,37]. Prolactin concentrations have only been assessed in one study on intermittent fasting [13]. In this trial by Harvie et al. [13], prolactin levels remained unchanged after 24 weeks of the 5:2 diet in premenopausal women with overweight and obesity. These results, although very limited, suggest that intermittent fasting may be safe for lactating females. In support of this, a recent trial by Gray et al. [38] studied the effect of the 5:2 diet on weight loss and gestational diabetes risk reduction in breastfeeding women. The study showed that intermittent fasting was safe in this population group, well tolerated, and lead to no adverse changes in milk production. These findings warrant confirmation by a study that specifically examines the safety and efficacy of intermittent fasting in breastfeeding women.

#### 2.1.6. Females: Summary of Findings

These preliminary findings suggest that fasting generally decreases androgens (i.e., testosterone and FAI) while increasing SHBG in premenopausal females with obesity. These results offer promise for the use of intermittent fasting in the treatment of hyperandrogenic conditions such as PCOS. Though it is important to note that these results were generally only observed when food consumption was limited to earlier in the day. Thus, premenopausal women may need to finish eating by 4 pm each day to observe these benefits in androgen markers. On the other hand, fasting does not appear to have any effect on other reproductive hormones such as estrogen, gonadotropins, and prolactin. More research is needed to confirm these findings.

### 2.2. Males: Effects of Intermittent Fasting on Reproductive Hormone Concentrations

#### 2.2.1. Testosterone

Testosterone is the major androgenic steroid hormone in adult males and is responsible for maintaining sperm production, libido, and sexual efficacy [39]. Another key role of testosterone is to stimulate muscle protein synthesis, thereby increasing muscle mass.

The effects of fasting on testosterone levels have been examined in four clinical trials to date [8,10,11,12] (Table 2). All studies employed 8 h TRE as the fasting intervention, and most combined TRE with resistance training (three times per week). Each trial was conducted in physically active, lean, young men. The findings revealed that TRE alone and combined with resistance training consistently reduced total testosterone levels [8,10,12] and free testosterone [11] after 4 to 44 weeks of intervention. The reductions in testosterone did not appear to be related to the duration of intervention, as shorter trials produced similar changes in testosterone as longer trials [8,10,11,12]. Mild weight loss was noted in each trial (1–3% from baseline) [8,10,11,12], and the degree of weight loss did not seem to be related to the magnitude of the testosterone reduction. Body composition changes were also evaluated. In each trial, fat-free mass remained unchanged, while fat mass was reduced [8,10,11,12]. Interestingly, the decreases in the anabolic hormone, testosterone, did not lead to any deleterious body composition changes or compromises in muscular strength.

#### 2.2.2. SHBG

SHBG is a glycoprotein that serves as a major carrier of testosterone in the circulation—carrying 40–45% of bound testosterone—and, thus, serum testosterone levels are higher when SHBG levels are higher [40,41,42]. Obesity can directly contribute to lower testosterone levels in males by reducing levels of SHBG [43,44].

Only one fasting study [11] has measured SHBG levels in male participants. After 4 weeks of 8 h TRE alone, circulating concentrations of SHBG remained unchanged, but free testosterone was reduced [11]. Less free testosterone would suggest that more testosterone is bound by carriers, but surprisingly, no subsequent increases in SHBG were observed [11]. This finding suggests that total testosterone may have decreased or that testosterone became bound to other carriers such as albumin [40]. The short time frame of the study (4 weeks) may have also prevented meaningful changes in SHBG from being observed. Taken together, TRE can decrease testosterone levels among healthy, active males, without affecting SHBG levels.

### 2.3. Males: Summary of Findings

These findings suggest that TRE reduces free and total testosterone levels in lean, physically active, young men. SHBG, however, does not seem to change with TRE, though the findings are very limited. Interestingly, muscle mass and muscular strength were not negatively affected by the reduction in circulating testosterone levels.

## 3. Intermittent Fasting, the Gut Microbiome, and Sex Hormones

Studies have shown that alterations in the gut microbiome can significantly affect reproductive hormones [45]. Improving abnormal microbiomes may lead to better reproductive health outcomes among pre- and postmenopausal women [46]. Clinical and preclinical data have shown that intermittent fasting can improve the composition and diversity of the gut microflora [47,48]. Periods of fasting have also been shown to reduce gut permeability leading to blunted postprandial endotoxemia and systemic inflammation, which are typically elevated in obesity [47,48]. Various members of the microbial community in the gastrointestinal tract can utilize endogenous (host) substrates during fasting, resulting in the production of metabolites beneficial to the host such as butyrate, acetate, and mucin stimulants [48,49]. Thus, it is possible that the effect of intermittent fasting on reproductive hormones is mediated by changes in the gut microbiome.

## 4. Circadian Rhythmicity and Sex Hormones

It has been proposed that intermittent fasting impacts sex hormone levels via improved alignment of circadian rhythms [14,50]. Hormone production and release are, in part, controlled by circadian rhythms which are, in turn, influenced by daily feeding–fasting cycles [50]. It has been suggested that TRE shortens the daily eating window to better align with circadian biology, which may beneficially affect hormone levels [14,50]. As demonstrated by Jakubowicz et al. [14], there may be a benefit to timing caloric intake earlier in the day vs. later in regard to testosterone levels in women with PCOS. While these findings offer promise for the use of early TRE in improving certain sex hormone concentrations, further research is needed before solid conclusions can be reached.

## 5. Directions for Future Research

The evidence in this area is still very limited. Future studies in women should involve perimenopausal and postmenopausal females, since no studies have been performed in these groups of women to date. It will also be of interest to further explore the effect of different fasting regimens in women with PCOS. Given that TRE shows promise in treating hyperandrogenism in females with this condition, large well-powered RCTs will be needed to evaluate if fasting is indeed a viable treatment option. Future studies should also examine how other sex hormones in females respond to fasting such as progesterone and estrone. In males, it will be important to include those who are overweight or obese, as current studies have only involved athletic, healthy, lean males. Trials with longer duration (24–52 weeks) are also needed.

## 6. Limitations to the Current Body of Evidence

There are several limitations to the current body of evidence. First, very few studies have examined how intermittent fasting impacts sex hormones. This should be taken into consideration when interpreting the present findings. Second, these trials all had small sample sizes (*n* = 16–107) and measured reproductive hormones as secondary exploratory outcome measures. Thus, it is highly likely that none of these trials were adequately powered to detect statistically significant changes in any hormonal parameter. Third, many trials did not include a control group in their design. Thus, it is difficult to confirm if these results are due to the fasting intervention instead of other extraneous variables. Fourth, most of these trials were quite short; therefore, the long-term effects of intermittent fasting on reproductive hormones are still not known. Fifth, studies in women to date are limited to premenopausal females, while studies in men are limited to athletic, healthy, lean males. There are no studies in postmenopausal women or overweight/obese men, which greatly decreases the generalizability of our findings. Sixth, it is unclear if the day of the menstrual cycle was recorded and standardized for all the studies involving premenopausal females. Levels of LH, FSH, and estrogen vary considerably according to the day of menstruation and, thus, should be controlled for in future studies in this area.

## 7. Conclusions

In interpreting these findings, it is important to note that very few studies have been conducted in this topic area. Thus, it is difficult to draw solid conclusions at present. From the limited data presented here, it is possible that intermittent fasting may decrease androgen markers in both genders. If these findings are confirmed by future research, these results would have different health consequences for females and males. On the one hand, fasting may prove to be a valuable tool for treating hyperandrogenism in females with PCOS by improving menstruation, fertility, and quality of life. On the other hand, fasting may be shown to decrease androgens, which would be less desirable among males. Low testosterone levels can negatively affect metabolic health, muscle mass synthesis, and libido in males. Interestingly, there were no adverse changes in fat-free mass in response to this reduction in androgens. All other reproductive hormones remained unchanged for both genders in these short-term trials. While these findings provide some preliminary evidence, these data require confirmation by large-scale, well-powered RCTs designed to specifically examine how intermittent fasting impacts reproductive hormone levels in various population groups.

## Figures and Tables

**Table 1 nutrients-14-02343-t001:** Females: effect of intermittent fasting on reproductive hormones concentrations.

Study Design				% Change from Baseline								
Reference	Subjects	Duration (Weeks)	Interventions	BW	FM	FFM	Estradiol	Testosterone, AE	SHBG	FAI	DHEA	LH FSH PRL
**Time-Restricted Eating (TRE)**												
Li 2021 [9]	*n* = 18 Females Age: 18–31 y Overweight Obese Premenopausal PCOS	5	**Single arm** 1. 8 h TRE (8 a.m.–4 p.m.)	1. ↓2% *	--	--	--	Total T: 1. ↓9% *	1. ↑2% *	1. ↓26% *	--	1. LH: ∅ FSH: ∅
**The 5:2 Diet**												
Harvie 2011 [13]	*n* = 107 Females Age: 30–45 y Overweight Obese Premenopausal	24	**RT: Parallel arm** 1. 5:2 Diet: Fast day (500 kcal), Feast day (ad libitum) 2. CR (1500 kcal/d)	1. ↓7% * 2. ↓5% *	1. ↓ * 2. ↓ *	1. ↓* 2. ↓ *	--	Free T: 1. ∅ 2. ∅ AE: 1. ∅ 2. ∅	1. ↑14% * 2. ↑6%*	1. ↓6% * 2. ↓10% *	1. ∅ 2. ↓6% *	PRL: 1. ∅ 2. ∅
**Meal Timing**												
Jakubo-wicz 2013 [14]	*n* = 60 Females Age: 25–39 y Overweight Normal weight Premenopausal PCOS	12	**RT: Parallel arm** 1. >50% of daily calories consumed at breakfast 2. >50% of daily calories consumed at dinner	1. ∅ 2. ∅	--	--	1. ∅ 2.↑35% *	Free T: 1. ↓50% *† 2. ∅ Total T: 1. ↓47% *† 2. ∅ AE: 1. ↓34% *† 2. ∅	1.↑% *† 2. ∅	1.↓% *† 2. ∅	1.↓% *† 2. ∅	--

--: Not measured; ∅: nonsignificant change, ↓: decrease; ↑: increase. * *p* < 0.05, significantly different from baseline (within a group effect). † *p* < 0.05, significantly different from the control or comparison group (between group effect). AE: androstenedione; BW: body weight; CR: calorie restriction; DHEA-S: dehydroepiandrosterone sulfate; FAI: free androgen index (100 × (total testosterone/SHBG)); FFM: fat-free mass; FM: fat mass; FSH: follicle-stimulating hormone; LH: luteinizing hormone; PRL: prolactin; RT: randomized trial; SHBG: sex hormone-binding globulin; T: testosterone, TRE: time-restricted eating (prescribed eating window shown in parentheses); y: years.

**Table 2 nutrients-14-02343-t002:** Males: effect of intermittent fasting on reproductive hormone concentrations.

Study Design				% Change from Baseline				
Reference	Subjects	Duration (Weeks)	Interventions	BW	FM	FFM	Testosterone	SHBG
**Time-Restricted Eating (TRE)**								
Stratton 2020 [10]	*n* = 26 Males Age: 18–35 y Physically active	4	**RT: Parallel arm** 1.8 h TRE + CR25% + Resistance training 3×/week 2. CR 25% + Resistance training 3×/week	1. ↓1% * 2. ↓2% *	1. ↓9% * 2. ↓9% *	1. ∅ 2. ∅	Total T: 1. ↓1% * 2. ↓1% *	--
Moro 2020 [11]	*n* = 16 Males Age: 19 ± 2 y Elite cyclists	4	**RCT: Parallel arm** 1. 8 h TRE (10 a.m.–7 p.m.) 2. Usual diet (7 a.m.–9 p.m.)	1. ↓2% *† 2. ∅	1. ∅ 2. ∅	1. ∅ 2. ∅	Free T: 1. ↓27% *† 2. ↓8%†	1. ∅ 2. ∅
Moro 2016 [8]	*n* = 34 Males Age: 29 ± 4 y Resistance trained	8	**RCT: Parallel arm** 1. 8 h TRE (12 p.m.–8 p.m.) + Resistance training 3×/week 2. Usual diet (8 a.m.–8 p.m.) + Resistance training 3×/week	1. ↓ * 2. ∅	1. ↓15% *† 2. ∅	1. ∅ 2. ∅	Total T: 1. ↓21% *† 2. ∅	--
Moro 2021 [12]	*n* = 20 Males Age: 29 ± 4 y Resistance trained	44	**RCT: Parallel arm** 1. 8 h TRE (1 p.m.–8 p.m.) + Resistance training 3×/week 2. Usual diet (8 a.m.–8 p.m.) + Resistance training 3×/week	1. ↓3% *† 2. ↑3% *†	1. ↓12% * 2. ∅	1. ∅ 2. ↑3% *	Total T: 1. ↓17% * 2. ∅	--

--: Not measured; ∅: nonsignificant change. ↓: decrease; ↑: increase. * *p* < 0.05, significantly different from baseline (within group effect). † *p* < 0.05, significantly different from the control or comparison group (between group effect). BW: body weight; FFM: fat-free mass; FM: fat mass; RT: randomized trial; RCT: randomized control trial; SHBG: sex hormone-binding globulin; T: testosterone; TRE: time-restricted eating (prescribed eating window shown in parentheses).

## Data Availability

Not applicable.

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
