# Peer review of "Effect of Intermittent Fasting on Reproductive Hormone Levels in Females and Males: A Review of Human Trials"

_nutrients, 2022, doi:10.3390/nu14112343_

Round 1

Reviewer 1 Report

Comments and Suggestions for Authors

The article ‘Effect of intermittent fasting on reproductive hormone levels in females and males’ by Cienfuegos et al. provides a literature review on the effect of intermittent fasting on hormone levels and reproductive health in adult male and female humans. They performed a study on the literature in various online database and included primary results from randomized controlled trials and non-randomized trials. They describe and summarize the effect of intermittent fasting (IF) (ADF, 5:2, TRE) on levels of Estradiol, Androgens, SHBG, Gonadotropins and Prolactin. They provide their results separated for females and males and summarize their findings at the end.

In principal, this is a well-written manuscript of scientific interest but it is very short and some of the studies are only roughly described.

Comments to the authors:

Are there any explanations found why the hormone levels change in the studies? Did they analyses for human or microbiome derived metabolites that affect host metabolism and hormone production? In SHBG they describe that there was no reduction in body weight, but was a reduction in fat (as hormone producing tissue)?

As we know from other research (https://doi.org/10.1152/ajpgi.00475.2020) that IF has a major role on the gut microbiome it would be very interesting to add one paragraph in the changes microbiome, their metabolites and their potential role on host metabolism and hormone production.  

IF stimulates a rhythm what’s about the role of the Circadian rhythm and sex hormones. Is it really it really only the fasting stimulus or did some of the studies also treat the question of the circadian rhythm without changing nutritional aspects.  

Gonadotropin: in what dimension was the weight loss; was the study on obese women? Indicate that and describe a little bit more detailed what the findings of the studies were.

Line 188: space missing.

Line 201: are there really only studies on premenopausal women with PCOS available or did the authors focus on that topic? If they focused on it they should change the title.

Males: are there any studies on obese men available?

Author Response

Please see attached response.

Reviewer 2 Report

Despite the important topic discussed in the publication, I believe that the number of publications taking into account changes in the hormonal profile in humans after IF is insufficient to draw even preliminary conclusions. Additionally, one of the works (Jakubowicz et al.) is not about IF but the distribution of calories in different meals throughout the day. Taking into consideration the varied impact of different types of intermittent fasting (TRE, ADF, PF), small study groups, only a limited number of publications, and no unified research methodology including articles do not allow for the formulation of scientific conclusions. There are other insignificant errors, but there is no need to mention them in detail in the existing situation.

Author Response

Please see attached response.

Round 2

Reviewer 2 Report

In my opinion, the authors did not improve the manuscript in such a way that it could publish in Nutrients

Author Response

Comment from the Editor

I thank the authors for being responsive to the reviewer's comments/concerns. While I agree with Reviewer 2 that the small number of studies is a major limitation, I also agree with the authors that this is actually an important point and a starting point for future exploration. However, I feel that the authors should somewhat temper their conclusions due to this small number of studies and highlight this point in either the limitations or the conclusion.

Thank you for taking the time to review our revised manuscript. We agree with the comments from the Editor. In view of the small number of studies, we have now tempered our conclusions (lines 18-25, and 310-324). We have also highlighted the limited number of studies in the limitations section (lines 293-294). We hope our revisions make our paper suitable for publication in Nutrients.